# *Photuris lugubris* Female Fireflies Hunt Males of the Synchronous Firefly *Photinus palaciosi* (Coleoptera: Lampyridae)

**DOI:** 10.3390/insects13100915

**Published:** 2022-10-08

**Authors:** Yara Maquitico, Aldair Vergara, Ilana Villanueva, Jaime Camacho, Carlos Cordero

**Affiliations:** 1Posgrado en Ciencias Biologicas, Universidad Nacional Autonoma de Mexico, Ciudad Universitaria, Ciudad de México 04510, Mexico; 2Carrera de Biologia, Facultad de Estudios Superiores Iztacala, Universidad Nacional Autonoma de Mexico, Ciudad de México 54090, Mexico; 3Carrera de Biología, Facultad de Ciencias, Universidad Nacional Autonoma de Mexico, Ciudad Universitaria, Ciudad de México 04510, Mexico; 4Departamento de Ecologia Evolutiva, Instituto de Ecologia, Universidad Nacional Autonoma de Mexico, Ciudad Universitaria, Ciudad de Mexico 04510, Mexico

**Keywords:** bioluminescence, deceptive behaviour, predation, behavioural dimorphism, polyandry, Mexico

## Abstract

**Simple Summary:**

Adults of most fireflies do not feed, except for the females of several species of *Photuris*, which eat males of other firefly species. In most cases, female *Photuris* attract their prey by responding with glows or flashes to the male’s bioluminescent signals, as if they were potential mates. For this reason, female *Photuris* are also called *femmes fatales*. In this paper, we provide evidence that the females of *Photuris lugubris* are *femmes fatales* of the males of the firefly *Photinus palaciosi*. We present different types of field observations suggesting that the females of *P. lugubris* attract males of *P. palaciosi* by responding to their bioluminescent signals and that, as a consequence, sometimes these are captured and eaten. We demonstrate experimentally that male *P. lugubris* are not predators of *P. palaciosi*. We also present experimental evidence that females of *P. lugubris* mate with multiple males and discuss the potential implications of this observation for the switch between mating and hunting behaviour. *P. palaciosi* is a firefly that reaches very high densities of flashing males during the reproductive season and that, intermittently, they synchronize their flashes providing a magnificent show that in the last few years has been the focus of tourist activities. Thus, our study also adds to the knowledge on the natural history of a species of economic interest.

**Abstract:**

*Femmes fatales* (*Ff*) are female fireflies that hunt and feed on the males of other firefly species that they attract by responding with glows or flashes to their bioluminescent signals. Here, we present field observations demonstrating that *Photuris lugubris* females are *Ff* of male *Photinus palaciosi*, a synchronous firefly exploited as a tourist attraction in the mountains of central Mexico. We show that the hunting success of the *Ff* is low, as observed in previous studies, suggesting that the impact of predation on the prey population is low. We present experimental data showing sex-specific hunting behaviour, since only female *P. lugubris* fed on *P. palaciosi.* We also present experimental data showing that at least some female *P. lugubris* mate multiple times; we discuss the implications of this discovery for the switch between the mating and hunting modes of *Ff*. We discuss open questions, as well as the possible impact of *Ff* on tourism focused on synchronous fireflies.

## 1. Introduction

Deception has evolved multiple times in the context of antagonistic interactions between species [1,2,3,4]. Members of the “deceiving species” transmit misleading information (“fake signals”) to members of the “deceived species” which respond in a way that is detrimental for them, but beneficial for the deceiving species. The costs paid by the deceived species generate selection pressures that could start antagonistic coevolutionary races [5]. For example, many palatable insect species have evolved morphologies, odours and colour patterns that make them similar to toxic or dangerous species (Batesian mimicry) and help them deceive potential predators [1,2,3,4].

*Femmes fatales* (*Ff* hereafter) are female fireflies that hunt and feed on males of other firefly species [6]; the males of species with *Ff* have not been observed hunting or feeding on fireflies. Firefly *Ff* are a textbook example of deceptive behaviour [2,4,7,8]. *Ff* attract their male prey by responding with glows or flashes to their bioluminescent signals, as if they were potential mates. *Ff* behaviour has been observed exclusively in the subfamily Photurinae [9]. In North America, *Ff* have been observed in *Bicellonycha* [10] and in several (but not all) species of *Photuris* belonging to the group called Division II by Lloyd [9]. According to Lloyd [9], this behaviour is also present in other Neotropical genera of Photurinae. In the North American species studied, *Photuris* females mimic “with ‘some degree’ of refinement” [11] (p. 370) the bioluminescent responses of the females of their prey species (this is called aggressive mimicry), mainly members of *Pyractomena* and *Photinus* [9,11,12,13]. Although males of prey species can escape before or during the attack, the predatory females eat many of them [12] and acquire, besides nutrients, defensive steroids called lucibufagins that *Photuris* are unable to produce [14]. Lucibufagins protect adult *Photuris* from predatory spiders [13] and protect their eggs from insect predators [14].

*Photuris* females enter the *Ff* predatory mode after mating [15], frequently changing location from their own mating site to the mating site of their prey species during the daily mating period [9,10]. Nelson et al. [15] reported that *Photuris versicolor* virgin females respond almost exclusively to artificial flashes mimicking the bioluminescent signals produced by conspecific males, and ignoring artificial flashes mimicking those of their male prey (*Photinus macdermotti*). In contrast, once mated, *P. versicolor* females respond exclusively to artificial heterospecific male flashes, ignoring flashes mimicking those of males of their own species [15]. The posture of virgin and mated female *P. versicolor* during the daily mating period also differs, with mated females being apparently more alert and ready to attack: “erect posture, extended mandibles and elevated antennae” [15] (p. 629); a photograph of this posture is also on page 629 of this paper. However, it is not known if other species with *Ff* need only one mating to switch to the *Ff* mode, since fireflies of other genera (e. g. *Photinus*) mate multiple times with different males [16,17]. On the other hand, Eisner et al. [13] described experiments in which they offered *Photinus* males to virgin *P. versicolor* females, which quickly attacked and consumed the males.

According to Lloyd [9,11], aggressive mimicry seems to be very common in *Photuris Ff*. However, at least in theory, it is possible that *Ff* attract prey fireflies by simply answering male prey with the same bioluminescent response they use to answer their own courting males, even if this response is different to that of females of the prey species. The evolution of aggressive mimicry in *Ff* depends on the shape of the preference function of males, defined as the relation between the tendency to approach a signal and the “structure” (the intensity, duration and time intervals between flashes) of the signal. If males are very selective (i.e., if they only approach females emitting a narrowly species-specific bioluminescent signal), aggressive mimicry will be selected in *Ff*. On the contrary, if males of the prey firefly respond (approach) to a broad range of female signals, selection for aggressive mimicry will be null or very weak. Intermediate situations are possible, and factors such as the abundance of *Ff* (and, thus, the strength of the selective pressure exerted on prey fireflies) and the presence and abundance of other firefly species whose signals interfere with intraspecific communication in the prey species, will determine the position along this gradient of a particular species.

In this paper, we present field observations and experimental data showing that *Photuris lugubris* females are *Ff* of male *Photinus palaciosi*, a synchronous firefly exploited as a tourist attraction in Mexico [18]. We also present data showing that at least some female *P. lugubris* mate multiple times before feeding on *P. palaciosi*. *Photinus palaciosi* is endemic to the mountains of Central Mexico [19], and its use in tourism is relatively recent [18]. To the best of our knowledge, this is the first report of an interaction between *P. palaciosi* and a species with *Ff*, and until now, it has been observed only in our study site (Rancho del Valle) near the town of Amecameca, Estado de México, México. For simplicity, we will refer to *P. lugubris* as “*Photuris*” and to *P. palaciosi* as “*Photinus*”.

## 2. Materials and Methods

### 2.1. Field Study

The study area is a pine-oak forest within a private property named “Rancho del Valle”, located in the Santiago Cuahutenco village, municipality of Amecameca, Estado de Mexico, Mexico. The main business of “Rancho del Valle” is tourism, including firefly sightings. The area of “Rancho del Valle” is 21 square kilometres, and although most of its area is occupied by forest, it has cabins, a restaurant, and other facilities. The fireflies were identified with the keys provided by Zaragoza-Caballero et al. [20] for the fireflies of Central Mexico.

Our observations and experiments were made during the reproductive seasons of two consecutive years: June 14th to July 21st in 2021, and May 18th to July 21st in 2022. In 2021 the reproductive season was already in progress when we started our observations, while in 2022 we started the study from the beginning of the season. Two to four observers per night walked along dirt paths between 20:00 and 24:00 h, a time interval that completely covers the nightly display period of *Photinus* and *Photuris* (~20:30–22:00 h). During the walks, we looked for female *Photuris* and once we located one, we recorded its activity. In some cases, we made focal observations of females, making voice or video recordings of their behaviour and interactions with *Photinus* or male *Photuris*. Most audio and video recordings were made with cell phones, although we also used a GoPro^®^ Hero 9 camera. Most of our observations were made in the dark, and are based on the luminescent signals emitted by the fireflies. Occasionally, we briefly used the light of a lantern or of a cell phone screen to check species or behaviours. Our observations were made mostly at a distance of at least one meter. The fireflies, perched either alone or interacting with other individuals never stopped their activities or moved away from the site when we observed them. For the experiments, we collected female *Photuris* and male *Photinus* (experiment 1) or *Photuris* couples and males (experiment 2), by hand or by using an entomological net. We took the collected specimens to the lab (less than 15 min away) in 125 mL plastic containers, one firefly per container.

### 2.2. Experiment 1: Do Male and Female Photuris Feed on Photinus Males?

In 2021, between June 18 and July 13, when female *Photuris* were frequently found in *Photinus* display areas, we captured *Photuris* females (N = 16) and males (N = 10) and kept them individually in ~1 litre empty plastic containers. The same night they were captured, we introduced one field-captured *Photinus* male to each of the containers. We maintained the containers in the dark and inspected them briefly and intermittently under a dim light throughout the night or until the *Photinus* male was eaten. We exposed all experimental *Photuris* just one night to a *Photinus* male and returned the live specimens to the field site after the experiment.

### 2.3. Experiment 2: Do Female Photuris Mate Multiple Times?

In 2022, we captured fourteen *Photuris* females at the beginning of their mating season (between May 28th and June 8th), twelve in copula and two when courting with a male (these couples mated in captivity the night they were captured); some of these females could have mated before. We kept the females individually in 1.1 litre plastic containers with a layer of soil—collected from the study site—in the bottom and ¼ of a small apple as a source of liquids. On consecutive nights, we exposed the females individually to a new male *Photuris*, also captured in the field. We observed the couples briefly and intermittently, under a dim light, until 1–1:30 am—three or four hours after the natural mating period—or before if they mated.

## 3. Results

### 3.1. Photuris lugubris Females Are Femmes Fatales of Photinus palaciosi Males

We made several field observations supporting the hypothesis that *Photuris* females are *Ff* of *Photinus* males. First, having studied *Photinus* during several years in other localities, mainly in the state of Tlaxcala, in our first year of study (2021) in Rancho del Valle, we did not expect to find *Photuris* in our study area. In fact, a recent article synthesizing information about the firefly species known from México and their distribution within the country did not report *Photuris lugubris* from the Estado de México [19], the state where our study site is located. However, in the first year of study (2021), we observed male and female *Photuris* in their mating display area, and later in the season, mainly females in or near *Photinus* display areas (see Section 3.2). Before knowing that it was a *Photuris* species, on at least 11 occasions we confused a *Photuris* female for a *Photinus* female in display areas of the last species.

Secondly, we observed several *Photuris* females perching on plants at different heights (from low grasses to leaves at 2 m in height) attracting and interacting at close range with *Photinus* males. We recorded this on 20 occasions (8 females in 2021 and 12 in 2022). In some of these cases, we observed *Photuris* females emitting glows (faint flashes that extinguished gradually) or flashes in response to (i.e., in the direction of) the male *Photinus* passing flashing, sometimes at distances as far as 2 m; when emitting flashes, females sometimes put their lanterns close to the leaf they were perching on, thus reducing the amount of light beholders could perceive. Some male *Photinus* responded by flying in the direction of the female. Some of these males landed on plants near the female (≤50 cm) and started flashing interactions with her, while other males stayed flying around the female, sometimes as close as 30 cm. In other cases, we observed *Photuris* females when they were already interacting with one or more *Photinus* males that were perched on nearby plants or flying around the female. During these interactions males emit flashes that the female responds with glows and flashes, however, not all *Photinus* flashes are responded to. In some interactions, the males came gradually closer to the female, usually walking or “jumping” on plants. We also observed females “jumping” on males when they were very close. Interestingly, males frequently “jumped down” when the female approached, or jumped in their direction, when they were a few cm from the female. The interactions between a female *Photuris* and an individual male *Photinus* can last just a few instants, but sometimes they extend for several minutes (our longest observation was up to 40 min). According to our observations, many (most?) of these interactions were unsuccessful hunting events. We have two clear examples of this in two females that interacted with several males in a relatively short period of time (our observations suggest that the rate of interaction varies broadly): (a) During 21 min of continuous observation, one female *Photuris* interacted with six *Photinus* males without success, even though she approached two of the males with her lantern off when they were close to her (these males jumped down from their perches before being captured); (b) During 36 min of continuous observation, one female *Photuris* interacted with five *Photinus* males unsuccessfully, although one of them barely escaped.

Thirdly, we observed five *Photuris* females (two in 2021 and three in 2022) that had already captured and were feeding on a male *Photinus*. Female *Photuris* frequently emit light when feeding on *Photinus*; we never observed *Photinus* flashing when attacked by *Photuris* (in contrast, males flash continuously when wrapped in spiderwebs). Finally, we observed four predation events from the moment the male approached the female until he was captured and eaten (two in 2021 and two in 2022), and one event (in 2021) in which the female captured the male and then lost him. As mentioned above, the female approached the male with her lantern off and captured the males, sometimes jumping on them.

### 3.2. Hunting Photuris lugubris Females Move to the Display Areas of Photinus palaciosi

The annual mating season of *Photuris* starts and finishes earlier than that of *Photinus*, although there is some overlap (personal observation). Our observations indicate that *Photuris* males emerge a few days before females (i.e., they are protandrous). In 2022, the first males were observed on May 12th, while the first females were observed on May 28th. At the beginning of the mating season, *Photuris* courts (i.e., perform bioluminescent displays and male–female interactions) exclusively on a slightly sloped area of a little less than 1 ha, where pine trees were planted more than 20 years ago; despite their age, these pines are small (<2m tall). When the mating season of *Photinus* begins, *Photuris* females start appearing in or near the display areas of *Photinus* (the first hunting female was observed on 9 June), sending bioluminescent responses to the displaying heterospecific males. *Photinus* also appears to be protandrous (personal observation). At the beginning of its mating season, *Photinus* mainly courts in a few, localized display areas (that do not coincide with the pine plantation display area of *Photuris*), and then later in the season, the display area expands to different parts of the forest. Accordingly, *Photuris* females are also present in these display zones at these times.

### 3.3. Only Female Photuris lugubris Eat Photinus palaciosi Males

None of the *Photuris* males confined with a *Photinus* male for one night (N = 10) killed or ate the *Photinus*, and although we only inspected them intermittently, they were never observed attempting to attack (Figure 1). In contrast, 15 out of 16 *Photuris* females (~94%) attacked the *Photinus* male and 13 of them ate the male (Figure 1 and Figure 2); one female attacked and then left the male alive—possibly because he faked death—and another female captured and then lost the male due to human disturbance. Although we only intermittently inspected the female containers, in several cases we observed *Photinus* males forcefully trying to escape, sometimes succeeding. However, eventually most were eaten, probably because they were confined with the female for several hours.

### 3.4. At Least Some Photuris lugubris Females Mate Multiple Times

We exposed fourteen mated females to between three and six males in captivity. Six of these females (42.9%) mated a second time and two of these re-mated females mated multiple times (three and six times in total, respectively) (Figure 3). Besides the fact that some of the females could have mated before we captured them for the experiment, a significant, positive correlation between the number of nights a female was exposed to a male and the number of copulations achieved (Spearman rank correlation: *r_s_* = 0.65, *p* = 0.011, *n* = 14; Figure 3) suggests that we underestimated both the proportion of re-mated females and the number of matings per female.

## 4. Discussion

Our observations show that female *Photuris lugubris* prey on *Photinus palaciosi* males in our study site. Female *Photuris* attract males by responding with glows and flashes to the bioluminescent signals of male *Photinus*, and then try to capture them once they are close. So, the behaviour of female *Photuris* fits the definition of a *Ff* [6,21]. Published discussions of *Ff* biology, e.g., [6,11,12,21], appear to suggest that species with *Ff* exhibit aggressive mimicry, however, as discussed in the introduction, this is not a necessary condition. We have not made a quantitative comparison of the bioluminescent signals displayed during courtship by *Photinus* females with those produced by *Photuris* females when courting and hunting, and thus we still do not know if the *Ff* of our study exhibit aggressive mimicry.

Although the hunting success of *Photuris* females in captivity was high—probably because they were confined in a relatively small space—our preliminary field observations suggest that the hunting success of *Photuris* females is relatively low, as we attested more failed than successful interactions. In most cases in which *Photinus* males interacted with a *Photuris* female while in flight or perched on a plant, either they never approached close enough to be attacked or they escaped by dropping from their perch when the females approached them. Our observations qualitatively agree with previous studies, estimating that between 10% and 15% of hunting attempts result in prey capture [12,22].

Our experimental results show that only female *Photuris* prey on *Photinus* males, as neither experimental (Figure 1) nor field-observed males attacked or fed on male *Photinus*. Although Eisner and collaborators have shown that by feeding on *Photinus Ff* obtain a type of defensive steroid (lucibufagins) that protects the female and their eggs from several predators [13,14], our observations indicate that *Photuris* consumes most of the soft parts of the body of their prey and thereby should also obtain energy and other nutrients from *Photinus* males. There are few studies on the number of prey eaten by *Ff* in the wild (according to Lewis [21], females of some species are capable of eating several males per night, and Lewis et al. [23] report that one female *Photuris* ate eight out of 11 *Photinus* males offered in captivity over a period of seven days), and we have not found any publication on the effect of the variation in the number of prey eaten on different components of their reproductive success. Why males do not feed on *Photinus* males is also an intriguing question that remains to be answered.

A previous study suggests that females are either in a sexually receptive state or in hunting mode [15]. However, if females mate multiple times and capture multiple prey, it is possible that they experience an intermediate phase in which they are still sexually receptive but are already trying to feed on *Photinus* males. We do not know if in our study site female *Photuris* attempted to eat several *Photinus* males, but we know that at least some females mated multiple times (Figure 3). We also observed one female *Photuris* courting with a male that on several occasions responded to the signals emitted by *Photinus* males flying nearby; the signals employed by the female while courting and when responding to *Photinus* males were clearly different.

Finally, measuring the impact of *Photuris* predation on *Photinus* demography is an interesting question for evolutionary and applied reasons. Lloyd [11] (p. 370) considers that predation by *Photuris* females is “probably one of the most important selection pressures affecting firefly signalling behaviour in the Western Hemisphere”. On the other hand, *Photinus palaciosi* is a synchronous firefly that is the focus of growing tourism activities in Central Mexico [18,24]. Since the success of firefly-watching tourism depends strongly on the large numbers of flashing fireflies, typically of synchronous fireflies, one could ask about the risks of severe population declines resulting from *Photuris* predation, both in our study area (the only site known to date in which *Ff* attacking *P. palaciosi* has been observed) or in other *P. palaciosi* populations where *Photuris* could exist or invade. The large numbers of *Photinus* males relative to the number of *Photuris* females observed in our study site, together with the apparently low hunting success, suggest that under the present conditions, the effect on *Photinus* mortality is small. However, the quantitative investigation of the demographic effects of *Ff* on their prey and modelling studies of different scenarios (for example, a population explosion of *Photuris*) seem worth pursuing.

## Figures and Tables

**Figure 1 insects-13-00915-f001:**
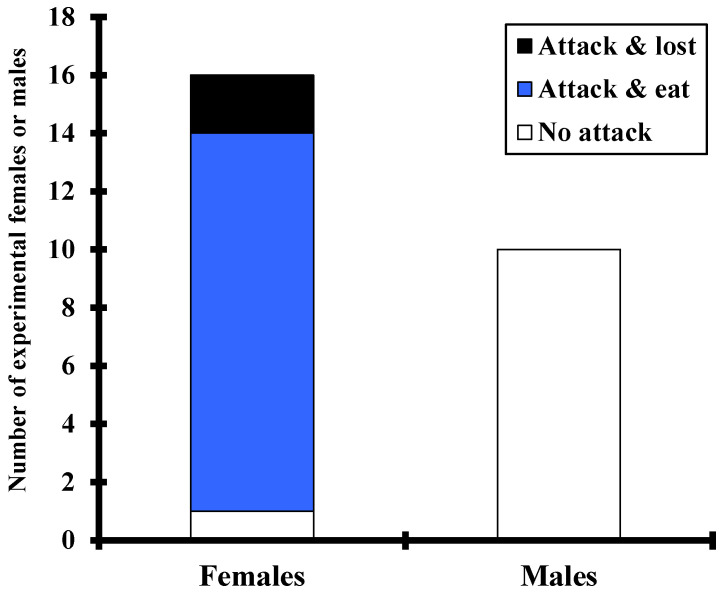
Number of experimental, field-collected *Photuris lugubris* females and males that did not attack, attack-and-ate or attack-and-lost the male *Photinus palaciosi* introduced to their containers.

**Figure 2 insects-13-00915-f002:**
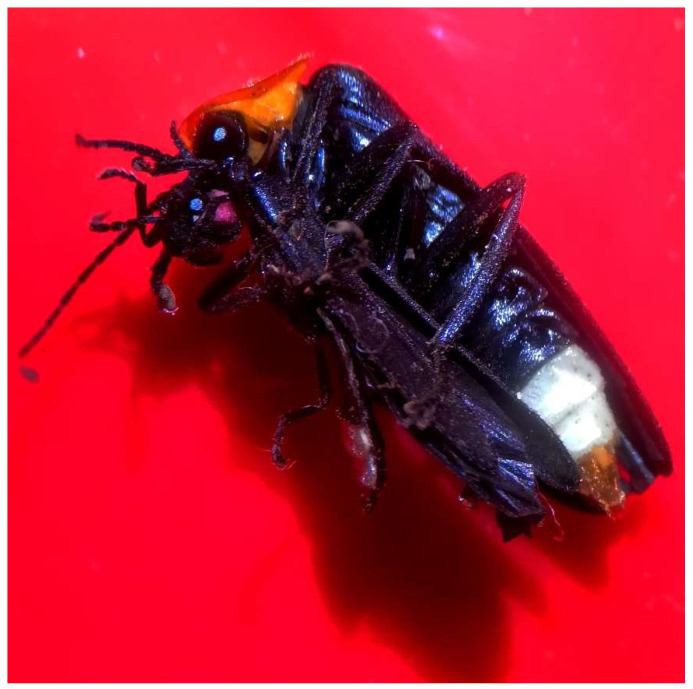
Experimental female *Photuris lugubris* (right) feeding on a male *Photinus palaciosi* (left). Notice that the prey is almost decapitated.

**Figure 3 insects-13-00915-f003:**
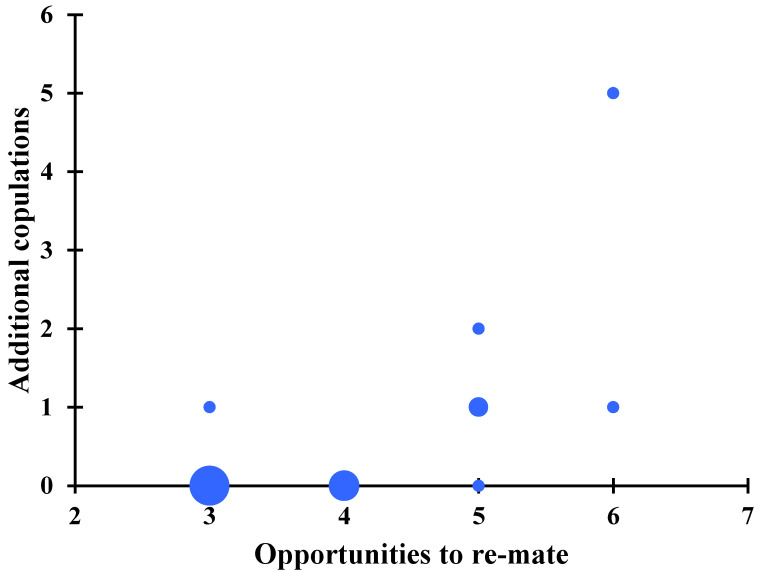
Number of copulations performed in captivity by females collected in copula, as a function of the number of opportunities to re-mate given in captivity. The positive correlation is statistically significant (Spearman rank correlation: *r_s_* = 0.65, *p* = 0.011, *n* = 14). The size of the blue dots indicates the number of females (smallest dot: one female; largest dot: four females).

## Data Availability

All relevant data are included in the manuscript.

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
