# Peer review of "Photuris lugubris* Female Fireflies Hunt Males of the Synchronous Firefly *Photinus palaciosi* (Coleoptera: Lampyridae)"

_insects, 2022, doi:10.3390/insects13100915_

Round 1

Reviewer 1 Report

Here, Maquitico and colleagues describe natural history observations of native Mexican fireflies, including the predatory behavior of Photuris lugubris on Photinus palaciosi.

The manuscript is very well written, easy to read and to understand. The introduction is well researched. These types of natural history observations are precious & this is definitely a good example of a well done work. I would recommend the manuscript be published, barring some minor comments:

(1) The fireflies described in this manuscript are identified to the species level, however the manuscript does not mention how this identification was performed (i.e., inference from dissected samples back to individuals in the field, flash patterns, etc.) . The identification methodology should be included, and any uncertainty in the identification, if there is any, should be stated.

(2) For some of the described behaviors “jumping” “jumping down”, it would be nice to have supplementary videos that demonstrate what the authors mean. This is completely optional; I leave it up to the authors.

Author Response

We thank the reviewer for their kind commentaries. With respect to their two specific commentaries:

“(1) The fireflies described in this manuscript are identified to the species level, however the manuscript does not mention how this identification was performed (i.e., inference from dissected samples back to individuals in the field, flash patterns, etc.) . The identification methodology should be included, and any uncertainty in the identification, if there is any, should be stated.”

At the end of the first paragraph of section 2.1 of the revised version of our manuscript (RV), we mention that the fireflies were identified by following the keys for the fireflies of Central Mexico, recently published by Zaragoza-Caballero et al. (2020).

“(2) For some of the described behaviors “jumping” “jumping down”, it would be nice to have supplementary videos that demonstrate what the authors mean. This is completely optional; I leave it up to the authors.”

Unfortunately, since we tried to reduce as much as possible the effect of our presence, we used voice recordings or video recording without light, and we do not have any good video of this behaviour.

Reviewer 2 Report

Interesting manuscript. I think the title needs to be changed since it is sensationalizing the fact that thes Ff could be bad for tourism but you results show that that is not the case and predation rate is way too low to have an impact. You also talk about this only in the last paragraph so it is really not as important a part of the manuscript as the title suggests.

Otherwise a nice manuscript which only needs a few changes.

Author Response

We thank the reviewer for their kind commentaries. Regarding their specific suggestion about the title:

“Interesting manuscript. I think the title needs to be changed since it is sensationalizing the fact that thes Ff could be bad for tourism but you results show that that is not the case and predation rate is way too low to have an impact. You also talk about this only in the last paragraph so it is really not as important a part of the manuscript as the title suggests.”

We agree that the title could be somewhat misleading and have decided to follow their suggestion about changing the title, by eliminating the first sentence (Femmes fatales feeding on a tourist attraction).

We also made all the changes suggested by the reviewer in the original manuscript, with exception of changing litre for liter, because we are trying to follow the british spelling (as in behaviour).

Reviewer 3 Report

Very interesting work describing hitherto unknown relationships between two species of Lampyridae. Field observations and experiments carried out by the Authors made it possible to better understand the biology of these interesting beetles.

The paper was written very carefully, minor editorial errors are as follows:

Page 8, line 258

- redundant dash after „space”

Page 10 “References” section

- double numbering of all references

- line 329 – the title should be in italics

- line 330 – the title should be in italics

- line 342 – redundant space before “PNAS”

Author Response

We thank the reviewer for their kind commentaries. We have made all the corrections suggested in their “minor editorial errors” list.

Reviewer 4 Report

Dear Authors,

The manuscript titled "Femmes fatales feeding on a tourist attraction: Photuris lugubris females hunt males of the synchronous firefly Photinus palaciosi (Coleoptera: Lampyridae)" by Maquitico et al., is an interesting read, but I am afraid to say that the study does not sound scientifically correct. The entire study lacks experimental planning, right hypothesis and incorrect execution of the experiments. 

The introduction itself is written is written like an abstract. None of the insect behaviours were clearly explained, be it the "response" of Photuris female to the calling behavior of Photinus males. I wonder what you meant by 'with come degree of refinement' ? The way the introduction is written, I had many questions about the behavior, but none of answered clearly. if Lucibufagins (seriods) provide protection in Photuris females against predation from spiders, but this compound doesn't provide protection to Photunis males ?

Results: The numbers are too low to draw any conclusion. I only see the observations but not results in the 'results' section. I did not find any reason why I had to read discussion! 

Author Response

We thank the reviewer for their specific commentaries that we considered in the following way:

“None of the insect behaviours were clearly explained, be it the "response" of Photuris female to the calling behavior of Photinus males.”

Although in the original manuscript we explained the behavioural responses of female Photuris in the Introduction (e.g. second paragraph) and Results (e.g. second paragraph of subsection 3.1), we did not mention it in the abstract and in some other parts of the text. In the revised text, we have made explicit that female Photuris respond with glows or flashes to the luminic signals of male Photinus almost every time we mention the response of female Photuris.

“I wonder what you meant by 'with come degree of refinement' ?”

As indicated in the original manuscript, this is a verbatim quotation of Lloyd (1984). Some degree of refinement means that the deceiving signals of the femmes fatales of the different species of Photuris show different degrees of similarity to the signals of the females of their prey species.

“if Lucibufagins (seriods) provide protection in Photuris females against predation from spiders, but this compound doesn't provide protection to Photunis males ?”

In the Introduction we described observations and experiments previously done. Why male Photuris in all species studied so far does not feed on fireflies with lucibufagins is the big (unanswered) question (that we mentioned in the original manuscript at the end of the third paragraph of the Discussion). By the way, as far as we know, our study is the first experimental test of the sexual dimorphism in hunting behaviour observed in Photuris; all previous information appears to be based in the absence of field observations of male hunting.

Round 2

Reviewer 4 Report

L45: “them” refers to the species that is deceived here but using such words in the scientific writing should come with caution. This could lead to confusion!

L60-61: The word “them” is a relative term, and statement sounds speculative. Please rephrase the sentence. The readers are interested in understanding the mechanism rather than numbers. The authors could elaborate on the behavior, how many species’ males are eaten, behavior of the responding females etc., to make it interesting.

L65-66: I do not see the connection between the first and second part of the sentence. 

L65: Does Ff female eat conspecific males too?

L67-69: Please rephrase: Nelson et al. [15] report that virgin Photuris versicolor females respond almost exclusively to artificial flashes simulating those produced by males of their own species, ignoring artificial flashes simulating those of their male prey fireflies (For example: “Nelson et al reported that Photuris vercicolorvirgin females respond exclusively to artificial flashes mimicking the bioluminescent signals produced by conspecific males compared to the signals that mimic their male preys .”)

L72: “The posture of virgin and mated female P. versicolor also differs” When? Does it mean that after mating, the females remain in that ‘posture’ for the rest of their life? Does the behavior changes when they encounter the potential host? Or while sending back the ‘signals’? Please elaborate the behavior in virgin and mated females! In addition, I understand explaining the insect behavior is very complex, but terms, such as “erect posture, extended mandibles and elevated antennae” do not help in understanding the offensive behavior in Photuris females, unless you explain the behavior when the females encounter conspecific males during mating. 

L79: I absolutely do not see the significance of figure 1 here. Although, the authors tried to establish the concept in their introduction that Ff respond to courtship signals. Here they postulate a different mechanism involved in preying. B1: How is it possible that hunting responses are same as courtship response?  What is “hunting response” and “courtship response” ? To conclude: The introduction and the figure 1 are contradictory, and this figure do not add any value.

L84-86: The theory put forward in the figure 1 nullifies the finding by Nelson et al., 1975. Please see introduction (Line: 67-71). Again, the hypothesis is very speculative, and is not supported by previous literature.

L89: What is the structure here? In the entire manuscript, the authors did not specify from how far (distance) the females respond to male signals? 

L113-114: Please rephrase this sentence. Do not sound scientific (“a third species is common”).

The following sentence needs to be rephrased!!! L121-123: Two to four “trained” (trained for what?) observers per night “walked along dirt paths” (is this important in the study?) where the “two species of interest were present” (both species are geographically located in the same place?), “between 20:00 and 24:00 h” (move this part of the sentence after “dirt paths”), a time interval that completely covers the nightly display 122 period of Photinus and Photuris (~20:30–22:00 h).

L124: How was the female identification performed? (“and once we located one”). Was the data from “we recorded its location, time, and activity” used for the analysis? It is very likely that fireflies were ‘disturbed’ in their habitat and making audio/ video could have affected their natural behavior (?). Please specify how these observations were performed without interfering female behavior. Most insects respond to human-produced CO2, and it is important that authors clarify how all the observations were peformed. 

L125: Is this data used in the study / analysis? 

L129: couples of which species? 

L130: Whether all individuals were kept in the same container or different containers?

L132: It is an established fact that females fireflies feed on Photinus males.

L133-136: Between June 18 and July 13, the authors collected 16 females and 10 males of Photuris sp. The authors also mention that they collected males of Photinus sp too, which suggests to me that the authors had more than 26 Photinus individuals. Before I read into results, I would like to bring it to the authors notice that, they do not know how old (age) are the individuals (both species) they collected from the field? The authors do not know if the females / males of Photuris sp had already eaten males of Photinus before the individuals were collected by the observers and brought to the laboratory. Such experiments must be conducted by i) controlling the age of the individuals, ii) repeating the experiments with the individuals collected from different locations iii) testing the individuals for the fixed amount of time in order to understand predatory behavior in females, iv) reporting other behavior(s) displayed by the female fireflies when they encounter male fireflies of Photinus sp., v) conducting the experiments under ‘similar’ climatic conditions (temperature and relative humidity for example).

L141: “2.3. Experiment 2: Do Female Photuris Mate Multiple Times?” and L144: “some of these females could have mated before.”: If the objective was to access the mating frequency in Photuris sp., and if some of these female fireflies were already mated, how is it possible to test this hypothesis in the first place? In a scenario, for example, if they mate in the lab conditions, it is very likely that they are mating for the 2/3rd time. In another scenario, if the females do not mate in the laboratory, it is also very likely that these individuals were already mated. The right approach should have been dissecting 50% of the females to look for developing eggs. Another approach should have been that these females were kept/ reared in the laboratory for the remaining period and see if the females lay eggs. 

L146-147: Apple as a source of liquids? The simple solution would have been 5-10 % sugar solution. This is very standard procedure when doing experiments under control conditions.

 “We made several field observations supporting the hypothesis that Photuris females 153 are Ff of Photinus males” So, where are those results. I would like to see numbers and analysis of those numbers. L153-163: Results section generally have ‘results’. I do not see any results in this section.

L184: “According to our observations, many (most?) of these interactions are unsuccessful hunting events.” The number are missing!! To be scientifically sound, authors MUST generate number data to substantiate ANYTHING they write in the results section. 

I highly recommend that authors repeat their experiments next time, but make a video recordings of all the behavior they have mentioned in the results section. Until these experiments are performed, I am afraid these results are not results for me. I understand that authors have made enough efforts in making these observations, but there is absolutely no evidence to show to the readers. In addition, there is absolutely no protocol in collecting the data. If one has to repeat these experiments in the future, it is impossible to do so.

L221-222: “Photuris females (~94%) attacked the Photinus male and 13 of them ate him” and L184-185: “According to our observations, many (most?) of these interactions are unsuc-184 cessful hunting events”. Both these results are contradictory!!! Does authors have any explanation for this? Which of these results are more reliable and can be repeated? These things MUST be made clear!!

Figure 3 can only be supplementary, and it has no significance in the results section. The figure DO NOT have added value in the manuscript. 

L234-242: The number is extremely low to draw any scientific conclusions. These experiments / observations have to be made as replications 

I do not see any need to read through the discussion because, I am not satisfied with the results and overall manuscript.
